# Fatty Acids and Oxylipins as Antifungal and Anti-Mycotoxin Agents in Food: A Review

**DOI:** 10.3390/toxins13120852

**Published:** 2021-11-30

**Authors:** Mei Qiu, Yaling Wang, Lijun Sun, Qi Deng, Jian Zhao

**Affiliations:** 1College of Food Science and Technology, Guangdong Ocean University, Zhanjiang 524088, China; qiumei1@stu.gdou.edu.cn (M.Q.); sunlijun@stu.gdou.edu.cn (L.S.); dengqi@gdou.edu.cn (Q.D.); 2Guangdong Provincial Key Laboratory of Aquatic Product Processing and Safety, Zhanjiang 524088, China; 3Key Laboratory of Advanced Processing of Aquatic Products of Guangdong Higher Education Institution, Zhanjiang 524088, China; 4Guangdong Provincial Engineering Technology Research Center of Marine Food, Zhanjiang 524088, China; 5School of Chemical Engineering, The University of New South Wales, Sydney, NSW 2052, Australia; jian.zhao@unsw.edu.au

**Keywords:** fatty acids, oxylipins, antifungal, anti-mycotoxin

## Abstract

Fungal contamination of food, especially by mycotoxigenic fungi, not only reduces the quality of the food, but can also cause serious diseases, thus posing a major food safety challenge to humans. Apart from sound food control systems, there is also a continual need to explore antifungal agents that can inhibit fungal growth and mycotoxin production in food. Many types of fatty acids (FAs) and their oxidized derivatives, oxylipins, have been found to exhibit such effects. In this review, we provide an update on the most recent literature on the occurrence and formation of FAs and oxylipins in food, their effects on fungal growth and mycotoxin synthesis, as well as the genetic and molecular mechanisms of actions. Research gaps in the field and needs for further studies in order to realizing the potential of FAs and oxylipins as natural antifungal preservatives in food are also discussed.

## 1. Introduction

Fungal contamination of agricultural and food products is a major cause of food spoilage, which leads to food wastage and substantial economic losses [1]. Fungi are opportunistic organisms that can contaminate foods easily through their spores in the air if the hygiene of the food processing and storage environments is substandard or the air quality not properly controlled [2]. Fungal contamination of foods can also pose a serious health issue due to the production of mycotoxins by certain fungal species. Mycotoxins are secondary metabolites produced by several fungal genera, which can cause a range of adverse health effects on humans, including carcinogenesis, mutagenicity, nephrotoxicity, teratogenicity, neurotoxicity, and immunosuppression [3]. Preventing fungal growth and mycotoxin biosynthesis are important topics in food safety research.

There are a number of strategies, including physical, chemical, and biological methods, that can be used alone or in combination to prevent fungal growth and mycotoxin production [3]. However, most of these methods involve high equipment and high energy costs, or may reduce food quality to some extent, which limits their application in the food industry. Therefore, there is a continual need to explore new and more economical methods, such as natural antifungal agents, for the control of fungal growth and toxin production in foods. Oils (especially nano-essential oils) are new strategies to solve the above problems, not only for being of natural origin, but also because of having proven preservative efficacy against mycotoxin production. However, the high cost of essential oils and the controversy in biosafety have restricted its practical application [4,5,6]. In recent years, fatty acids (FAs) and their derivatives, oxylipins, have emerged as potential natural antifungal agents. FAs are important cellular molecules that are associated with cell membrane structure, energy storage, regulation of inflammatory processes, and various signaling pathways [7,8,9,10]. As components of fats and oils, they are widely present in many foods. A large number of FAs are found to have fungicidal or fungistatic properties and are used as antifungal agents in agriculture and medicines [11,12]. Oxylipins are a family of secondary metabolites derived from the oxidation or further transformation of polyunsaturated fatty acids (PUFAs). They are widely present in organisms from all the kingdoms of nature [13,14]. Oxylipins are found to have a strong impact on fungal development, reproduction, synthesis of secondary metabolites (including mycotoxins), and adaptive responses [14].

Here, we review the structure and formation of FAs and oxylipins in food, their inhibitory effects on fungal growth, and mycotoxin synthesis, as well as the mechanisms of their antimicrobial actions. We also discuss the challenges associated with the exploration of PUFAs and oxylipins as antifungal and anti-mycotoxin agents in the food industry and the research needed to address these challenges.

## 2. Effects of FAs on the Regulation of Fungal Growth and Mycotoxin Synthesis

Lipids are important physiological molecules related to cellular energy storage (e.g., adipose tissue), membrane structure (phospholipid bilayer), and various signaling pathways [15]. They are also found to be linked with mycotoxin production in foods. It is reported that when *A. parasiticus* contaminated crop seeds with different lipid contents such as peanut, paddy, sorghum, cowpea, and green gram, it produced the greatest amount of AFB_1_ in peanut, which has the highest level of lipids. Furthermore, *A. parasiticus* colonies were larger on powdered seed material than on the defatted samples of the same material [16]. Research has also shown that *A. flavus* and *A. parasiticus* produced greater amounts of aflatoxins on oil seeds or the lipid-rich tissues of seeds than on starch rich materials [17,18,19]. Similarly, Rajasekaran et al. [20] found that fungal spread and aflatoxin production in infected cottonseed were closely related to lipid accumulation in the seed, and the seed lipids played an important role in regulating the growth of *A. flavus* and *A. parasiticus* and the production of aflatoxins. In addition, *A. ochraceus* has been shown to infect peanuts and soybeans, and produce more ochratoxin A (OTA) on these high fat content seeds than on corn and wheat [21].

FAs can also regulate the growth and secondary metabolism of fungi; however, the effects are dependent on both the type and concentration of FAs. For example, Gupta et al. [22]. found that when lauric acid was added to a liquid medium for *A. parasitica*, aflatoxin synthesis by the fungus was significantly promoted, while myristic, palmitic, stearic, and oleic acids were less effective. Table 1 lists various FAs that have been shown to regulate fungal growth and mycotoxin production. In general, the antifungal efficiency of an FA is related to the structure and concentration of the FA. For example, the addition of 10 mM oleic acid promoted aflatoxin synthesis, while at 150 mM, it inhibited its synthesis in vitro. Furthermore, 50 mM oleic acid was found to promote *A. parasiticus* growth but inhibit aflatoxin synthesis; however, linoleic acid had the opposite effect at the same concentration [23].

## 3. Roles of Oxylipins in Regulating Fungal Growth and Mycotoxin Production

Oxylipins are a family of secondary metabolites derived from the oxidation or further transformation of polyunsaturated fatty acids (PUFAs). They are widely present in all organisms and play a significant role in their physiology [13,14]. For fungi, oxylipins are involved in the regulation of cell development, reproduction, synthesis of secondary metabolites (including mycotoxins), and adaptive responses [14]. Oxylipin can also be used as a natural preservative in food to control fungal growth and subsequent mycotoxin production. Lipidomics studies have shown that FA oxidation products, especially FA hydroperoxides, have a significant impact on the accumulation of fumonisin [29]. There is evidence that monohydroxy octadecenoic acid exerts antifungal activity in bread [30]. In particular, it has been shown that 13S-hydroperoxy-octadecadienoic acid (13S-HPODE) and 13S-hydroperoxy-octadecatrienoic acid (13S-HPOTE) inhibit aflatoxin production, while 9S-hydroperoxy-octadecadienoic acid (9S-HPODE) promotes it [31]. Similar to FAs, the effect of oxylipins on fungal growth and mycotoxin synthesis also depends on their structure and concentration (Table 2). Adding 9S-hydroxy-octadecadienoic acid (9S-HODE) to potato agar medium was found to inhibit the formation of conidia of *A. ochraceus* and promote the synthesis of OTA, while 13S-hydroxy-octadecadienoic acid (13S-HODE) at the same concentration had the opposite effect [21,32]. Another study found that mono- and di-unsaturated FAs with hydroxylation at position 9, 10, 12, or 13 exhibited similar fungal static activity, while unsaturated FAs with hydroxylation at position 2 or 18 appeared to have lower antifungal activity against *P. roqueforti* and *A. niger* [33]. Overall, the results of those study indicated that fungi and mycotoxins in foods can be controlled by adding oxylipin to foods or by modifying processing technology to regulate the formation of certain oxylipins in foods. The relationship between the structure and concentration of oxylipins and their antifungal activities provides an avenue for their application in foods for controlling fungal growth and mycotoxin production. However, the situation is complicated by the fact that a specific oxylipin may have different effects on different fungal species. For example, 13S-HPODE increases aflatoxin synthesis by *A. parasiticus* [34], whereas the same oxylipin decreases aflatoxin production by *F. verticillioides* [35]. This may prevent the selection of oxylipin with a wide spectrum antifungal application.

## 4. Pathways of Oxylipin Formation

Oxylipins, which are ubiquitous in all organisms, are the metabolites of PUFAs such as arachidonic acid (ARA), linoleic acid (LA), α-linolenic acid (ALA), eicosapentaenoic acid (EPA), and docosahexaenoic acid (DHA), formed by enzyme and non-enzyme oxidation of these FAs. Non-enzymatic pathways for the formation of oxylipins can occur through free radical-mediated oxidation of reactive oxygen species (ROS) [15,44,45]. ROS—mediated reaction can produce hydrogen peroxides from PUFAs, which are then rapidly reduced to a mixture of monohydroxy PUFAs [46]. The three main enzymes involved in their formation are lipoxygenase (LOX), cyclooxygenase (COX), and cytochrome P450 (CYP). LOX is reported to catalyze the formation of hydroperoxide from PUFAs with stereospecificity, and then convert them into monohydroxy PUFA (e.g., hydroxy-octadecadienoic, hydroxy-pentaenoic, and hydroxyl-docosahexaenoic acids), which can be further oxidized to corresponding ketones (e.g., keto-octadecadienoic acid, keto-eicosatetraenoic acid) [15,47,48]. CYP has ω-hydroxylation and cyclooxygenase activities. CYP cyclooxygenase catalyzes the formation of epoxide (e.g., epoxyeicosatrienoic, epoxy eicosatetraenoic, and epoxy docosapentaenoic acids), while CYP hydroxylase catalyzes the formation of medium-chain hydroxide and ω-hydroxylation (e.g., 20-hydroxypentaenoic and 18-hydroxyeicosapentaenoic acids) [15,47,49]. COX enzyme converts AA and EPA into peroxides hydroperoxide-endoperoxide prostaglandin G2 and peroxides hydroperoxide-endoperoxide prostaglandin G3 [15].

Plant oxylipins act as signals to modulate fungal developmental processes, including sporogenesis and biosynthesis of mycotoxins. These oxylipins are primarily derived from linolenic (C18:3 n-3), linoleic (C18:2 n-3), and hexadecatrienoic (C16:3) acids [50]. Over the last decade, understanding of the biosynthesis, metabolism, and regulation of plant oxylipins, especially jasmonate, has improved [13,50,51,52]. To some extent, plant oxylipins are recognized as part of the plant defense mechanisms against pathogenic attack. Compared with higher plants and mammals, the characteristics of fungal oxylipin synthesis are not yet comprehensively understood. Fungal oxylipin synthases, primarily including dioxygenases, lipoxygenase, and monooxygenase, are believed to mediate ω-oxidation, while multifunctional β-oxidase and hydroperoxide lyase are involved in the formation of volatile compounds. These oxylipin synthases catalyze C18 PUFA to form various compounds via multiple pathways in fungi (for a review see reference [14]).

## 5. Mechanisms Underlying the Antifungal Actions of FAs and Oxylipins

With regard to the antifungal mechanisms of FAs and oxylipins, it appears that the most important target of antifungal actions of FAs is the cell membrane (Figure 1). Avis and Bélange [53] reported that the general antifungal mechanism of FAs involves direct interactions with fungal cell membrane. Antifungal FAs insert themselves into the lipid bilayer of the fungal membrane and physically interfere with the membrane, resulting in an increase in the fluidity of the membrane. The increase in membrane fluidity causes general disruption of the cell membrane, which leads to conformational changes in membrane proteins, release of intracellular components, cytoplasmic disruption, and ultimately, cell disintegration. Vuyisile et al. [54] proposed that certain PUFAs increase the unsaturation index of the cellular membrane and accumulation of intracellular ROS in *C. albicans* and *C. dubliniensis*. These authors also found that C18:4 n-3, in particular, could cause apoptosis, probably by causing an increase in ROS production and loss of mitochondrial transmembrane potential (ΔΨm). Hydroxy FAs penetrate through fungal membrane bilayers to increase membrane permeability and release intracellular electrolytes and proteins [25,53,55]. However, monohydroxy unsaturated FAs with -OH in C9–C13 position inhibit the growth of the food spoilage fungi *P. roqueforti* and *A. niger*, indicating that fungal resistance to monohydroxy unsaturated fatty acids (HUFA) is related to the sterol content of the fungi biomass [33]. This suggests that food components that affect fungal sterol synthesis may enhance the antifungal activity of FAs and oxylipins.

PUFAs have also been shown to affect fungal reproduction. Fungal oxygenase catalyzed intracellular and extracellular PUFA to form oxylipins, which has been shown to play a significant role in regulating the growth of sexual and asexual fungal spores (conidia). In *A. nidulans*, the precocious sexual inducer (psi) factor is the first extracellular C18 PUFA derived oxylipin signal found to regulate spore development [56]. Psi-factor biosynthetic genes, named PPO (psi-producing oxygenases), are also involved in regulating fungal spore development and mycotoxin synthesis. Overexpression of ppoA results in a six-fold reduction in the ratio of asexual to sexual spore development [57]. Asexual development is significantly delayed and reduced in double ∆ppoA and ∆ppoC and triple ∆ppoA, ∆ppoB, and ∆ppoC mutants, accompanied by a premature increase in ascospore production [58]. In these processes, the light sensor protein VeA and the protein degradation machinery COP9 are involved in the regulation of the PPO genes [59,60]. The cleistothecial (NsdD)-specific and conidiophore (BrlA)-specific transcription factors regulate the sporulation program of the fungus [61,62] (Figure 2). In the IRT4 strain, in which the dioxygenase genes (ppoA, ppoB, ppoC and ppoD) and one lipoxygenase gene (loxA) are downregulated, *A. flavus* was reported to change from asexual development to sclerotium production, but aflatoxin synthesis increased [63,64], indicating that the bioactive lipoproteins produced by PPO and LOX enzymes play a key role in regulating spore production and mycotoxin biosynthesis. Another study reported that the lipoxygenase-deficient *A. ochraceous* produced lower levels of linoleic acid-derived 13S-HPODE and displayed remarkably lower ochratoxin production, delayed conidia formation, and increased sclerotia production [65]. Moreover, mutation in a *F. sporotrichioides* PPO orthologue was found to reduce T-2 toxin production [66]. These findings demonstrate that it is possible to control the growth of fungi and the synthesis of mycotoxins by regulating the activity of PPO and LOX or the concentrations of oxylipins.

Oxylipins may also influence fungal growth and mycotoxin synthesis through their effect on G protein (GTP hydrolyzing proteins). G protein signaling has been shown to regulate the synthesis of aflatoxin and its precursor, sterigmatocystin (ST). The Gα subunit FadA and its RGS FlbA are part of an adenylate cyclase/cyclic adenosine monophosphate (cAMP)/protein kinase A (PKA) (PkaA) pathway, which controls ST production in *A. nidulans* via transcriptional and posttranscriptional regulation of aflR [67,68,69]. In recent years, studies have shown that oxylipins, as ligands, are sensed by G-protein coupled receptors (GPCRs) to regulate physiological processes in *Aspergillus* [70,71,72]. Affeldt and colleagues [70] presented evidence indicating that oxylipins stimulate a burst in cAMP, a downstream event of GPCR activation in *A. nidulans*, and that the GPCR mutant ΔgprD prevents the cAMP accumulation. Therefore, G protein responds to oxylipin signaling molecules and activates the cAMP/PKA pathway to regulate the growth and secondary metabolism of *Aspergillus* (Figure 2). In addition to *Aspergillus*, GPCRs encoded by various fungi, including *Magnaporthe grisea*, *Cryptococcus neoformans*, *Neurospora crassa*, *Verticillium* spp., and *Trichoderma* spp., have also been identified [73], but it is not clear whether FAs and their oxygenates activate intracellular signaling pathways through GPCRs to regulate toxin synthesis. The G protein coupled signaling pathway could serve as an important target for controlling fungal growth and its secondary metabolites. For instance, oxylipin receptors could represent potentially rich targets for antifungal agent development.

## 6. Oxylipins in Foods

Measurement of lipid oxidation is a common quality control practice in the food industry for foods with a high level of fats or oil. Commonly used parameters include peroxide value (POV) and thiobarbituric acid reactive substances (TBARS), which represent the hydroperoxide value and the secondary oxidation products of lipids and fatty acids, respectively. However, neither of these measures provides precise information about specific oxidation products, including oxylipins. At present, there are only a few reports on the concentration of oxylipins, and the effects of different processing conditions on their formation in food. Mechanically deboned meat produces a large amount of 9,10,13-trihydroxy-11-octadecenoic acid (9,10,13-THODE) during storage, and 9,10,13-THODE can be used as a marker for the presence of mechanically deboned meat in various minced meat products [74]. During the processing of Chinese sausages, LOX-catalyzed lipid oxidation occurs mainly during the curing stage and the early stages of drying. As the oxidation process progresses, 3-HODE, 9-HODE, 9,10-dihydroxyoctadecenoic acid (9,10-DHODE), and 9,10,13-THODE levels continue to increase. The oxidation catalyzed by LOX is replaced by non-enzymatic oxidation in the later drying stage [75]. Heat treatment reduces the content of oxylipins in milk, such as the concentrations of linoleic acid derivatives 9,10-dihydroxyoctadecenoic acid (9,10-DiHOME), 13-HODE, and 9-HODE. Furthermore, the concentrations of 13-hydroxyoctadecatrienoic acid (13-HOTrE) and 9-HOTrE of α-linolenic acid after high-temperature short-time and ultra-high temperature treatments are lower than those in raw and pasteurized milk. Compared with POV and TBARS, the concentration of 12,13-DiHOME is reduced by 50% during the heat treatment of milk, which can be used as a sensitive marker of early oxidation of lipids [76]. Therefore, changes in the concentrations of certain oxylipins in food are more representative of the effect of processing conditions on lipid oxidation than POV and TBARS values.

Numerous studies support that oxylipins produced by plants or fungi are important molecules regulating fungal growth and secondary metabolism. Therefore, oxylipins formed in food can also be used as antifungal and anti-mycotoxin agents in the food system. For instance, Gao et al. [21] observed that the accumulation of 13S-HODE inhibited the production of OTA in soybean culture medium, but its mechanism is unclear. Understanding of the formation characteristics of oxylipins not only can provide a better understanding of the lipid oxidation process in food, but can also enable exploration of their application for the control of fungal growth and mycotoxin production in food. Furthermore, the actual application of oxylipins in foods requires a detailed understanding of its different antifungal and anti-mycotoxin mechanisms, as each oxylipin content may have different action sites.

## 7. Conclusions and Remarks

In this review, we summarized current literature on the effect of FAs and oxylipins on fungal growth and mycotoxin synthesis. Studies on their mechanisms of action indicate that cell membrane, specific enzymes, and metabolic pathways are the targets of these compounds. The antifungal and anti-mycotoxin activities of FAs and oxylipins may provide new strategies for the prevention and control of fungal growth and mycotoxin production in food. However, to date, there are only a limited number of studies that have explored their potential in food applications. Several research questions need to be addressed before such potential can be realized. First, many antifungal and anti-mycotoxin compounds have shown high activities in model systems, but their efficacy in complex food matrices are poor. Understanding of the formation characteristics of oxylipins is helpful to improve their efficacy in real food systems. Second, the antifungal and anti-mycotoxin properties of FAs and oxylipins are highly dependent on their concentrations. Research is needed to screen FA and oxylipins with wide spectrum antifungal and anti-mycotoxin activity. Third, the stability of highly reactive FAs and oxylipins in different foods and under various processing and storage conditions require further investigation. Some antifungal hydroxyl FAs have been found to be heat stable in bread making [30]. For this, nanoencapsulation improves efficacy and stability of FAs and oxylipins. Fourth, FAs and their derivatives are odorous and can affect the sensory properties of food; these needs to be taken into account when considering using these compounds in foods. Finally, there is a lack of knowledge regarding the toxicity characteristics of various oxylipins. The safety of these compounds must be ascertained by acute oral toxicity in mice before they can be used as food additives.

## Figures and Tables

**Figure 1 toxins-13-00852-f001:**
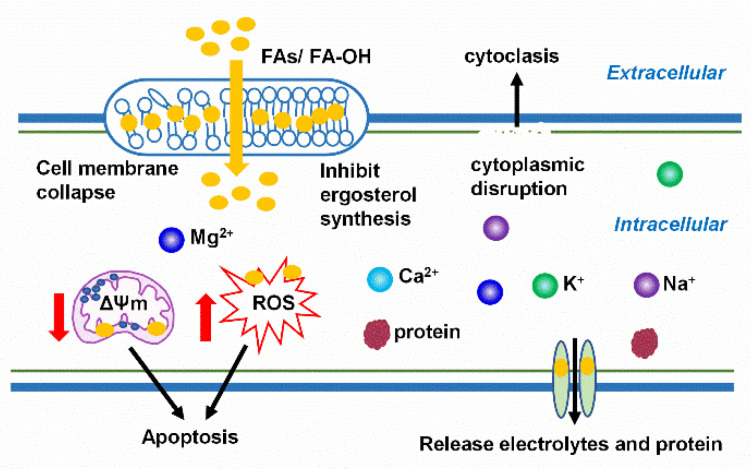
Antifungal mechanisms of free fatty acids and hydroxy fatty acids. Free fatty acids or hydroxy fatty acids (FAs/FA-OH) insert themselves into the lipid bilayer of the fungal membrane and result in general disruption of the cell membrane, low sterol content, release of intracellular components, cytoplasmic disruption, and ultimately, cell disintegration; they also cause an increase in ROS production and loss of mitochondrial transmembrane potential (ΔΨm) and ultimately, apoptosis.

**Figure 2 toxins-13-00852-f002:**
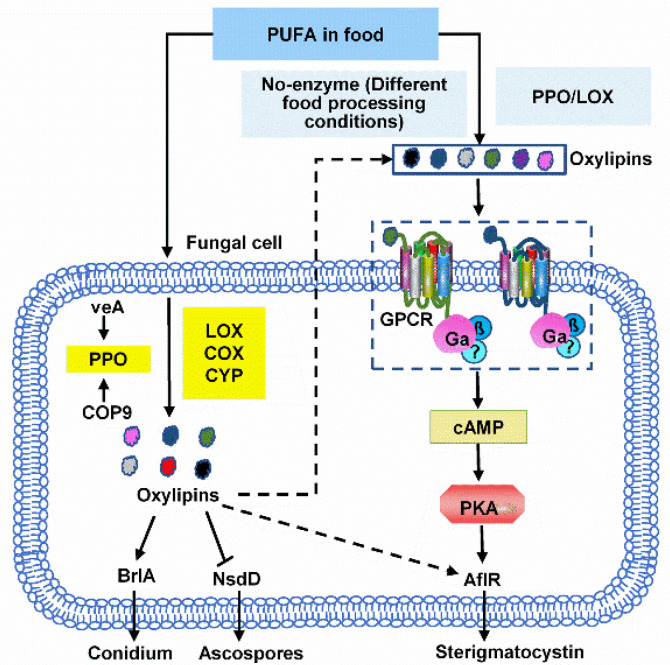
Hypothetical model of the regulation of fungal growth and toxin synthesis signal transmission in Aspergillus by oxylipins. Highlighted in yellow and blue are the fungal and food oxylipin biosynthetic enzymes. Polyunsaturated fatty acid substrates in food are processed by fungal-secreted lipoxygenase for oxylipin production. Fungi can sense and exploit oxylipins from food substrates to regulate GPCR-, cAMP/PKA- and Ppo-mediated growth, sporulation, and mycotoxin production. In these processes, the light sensor protein VeA and the protein degradation machinery COP9 are involved in the regulation of the PPO genes. Oxylipins regulate sporulation through cleistothecial (NsdD)- and conidiophore (BrlA)-specific transcription factors, as well as secondary metabolism (e.g., sterigmatocystin) through AflR.

**Table 1 toxins-13-00852-t001:** Effect of fatty acids in fungal development and mycotoxin production.

Fatty Acid	Concentration	Fungal Species	Fungal Growth	Mycotoxin Production	Ref.
lauric acid	50.00 mM	*Aspergillus parasiticus*	+	− AF	[23]
myristic acid			+	+ AF	
palmitic acid			−	− AF	
oleic acid			+	− AF	
linoleic acid			−	+ AF	
lauric acid	2.50 mM	*Aspergillus niger*	−	u	[24]
capric acid	0.60 mM	*Aspergillus fumigatus*	−	u	[25]
		*Aspergillus nidulans*	−	u	
	0.30 mM	*Penicillium commune*	−	u	
	0.15 mM	*Penicillium roqueforti*	−	u	
lauric acid	0.20 mM	*Aspergillus niger*	−	u	[26]
myristic acid	0.09 mM	*Penicillium glabrum*	−	u	
	0.13 mM	*Aspergillus niger*	+	u	
	0.13 mM	*Penicillium italicum*	−	u	
myristoleic acid	0.09 mM	*Aspergillus niger*	−	u	
	0.13 mM	*Penicillium italicum*	−	u	
palmitic acid	12.00 mM	*Botrytis cinerea*	−	u	[27]
stearic acid	0.10 mM	*Aspergillus flavus*	u	+ AF	[28]
linolenic acid	1.25 mM		u	− AF	

u: unknown; +: stimulate; −: inhibit; AF; aflatoxin.

**Table 2 toxins-13-00852-t002:** Proposed oxylipins with anti-fungal development and anti-mycotoxin production activities.

Qxylipin	Host Source	Fungal Species	Fungal Growth	Mycotoxin Production	Ref.
13S-HPODE	c.a	*Aspergillus flavus*; *Aspergillus parasiticus*	u	− AF	[34]
	peanut	*Aspergillus flavus*; *Fusarium verticillioides*	+	+ AF	[35]
13S-HODE	c.a	*Aspergillus ochraceus*	+	− OTA	[32]
9S-HPODE	soybean	*Aspergillus flavus*; *Aspergillus nidulans*	+/−	− AF/ST	[31]
	corn	*Aspergillus flavus*; *Fusarium verticillioides*	+	+ AF	[36]
9S-HOD(T)E	corn	*Fusarium verticillioides*	n	+ AFB_1_	[37]
9-HODE	wheat	*Fusarium graminearum*	u	+ DON	[38]
9-HPODE					
9S-HODE	c.a	*Aspergillus ochraceus*	−	+ OTA	[21]
9-HODE	maize	*Fusarium verticillioides*	u	+ FB_1_	[39]
10-HODE					
1-octen-3-ol	c.a	*Penicillium paneum*	−	u	[40]
	c.a	*Aspergillus nidulans*	−	u	[41]
	*Aspergillus oryzae*	*Aspergillus flavus*	−	+ AFB_1_	[42]
MeJA	c.a	*Aspergillus parasiticus*	−	− AF	[43]

c.a: commercially available; u: unknown; +: stimulate; −: inhibit; n: no effect; AF: aflatoxin; AFB_1_: aflatoxin B1; FB_1_: fumonisin B1; ST: sterigmatocystin; OTA: ochratoxin A; DON: deoxynivalenol. 13S-HPODE: 13S-hydroperoxy-octadecadienoic acid; 9S-HPODE: 9S-hydroperoxy-octadecadienoic acid; 13S-HODE: 13S-hydroxy-octadecadienoic acid; 9S-HOD(T)E: 9S-hydroxy-octadecadi(tri)enoic acid; 9-HODE: 9-Hydroxy-octadecadienoic acids; 10-HODE: 10-Hydroxy-octadecadienoic acids; 9-HPODE: 9-hydroperoxy-octadecadienoic acids; MeJA: methyl jasmonate.

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
