# Peer review of "Fatty Acids and Oxylipins as Antifungal and Anti-Mycotoxin Agents in Food: A Review"

_toxins, 2021, doi:10.3390/toxins13120852_

Round 1

Reviewer 1 Report

The manuscript "Fatty acids and oxylipins as antifungal and anti-mycotoxin agents in food: A review" is very interesting and well written and, in my opinion, fits the scoop of this journal. The text is well structured and provides a good insight about oxylipins as anti-mycotoxin agents and their mechanism of action. The text is easily comprehensible and has a very good English level.

I only have a few comments:

  • In table 1, convert the concentration values to the same unit (so it’s easier to compare values) also, the subtitle for the table is “…proposed function of fatty acids…” it would be better phrasing if you use “…effect of fatty acids…”
  • Some fungi names are not in italic.
  • In the section 6.Oxylipins in foods: you only describe oxylipins as a measure of lipid oxidation. I would like to see how the content of oxylipins in food affects fungal growth. Are there studies regarding this issue?

Author Response

Response 1: Thank you for your suggestion. In table 1, the concentration values have been converted to “mM” and subtitle for the table is “Effect of fatty acids…”

Response 2: The issues raised by the reviewer have been modified in the entire manuscript.

Response 3: Numerous studies support that oxylipins produced by plants or fungi are important molecules regulating fungal growth and secondary metabolism. Therefore, oxylipins formed in food can also be used as antifungal and anti-mycotoxin agents in the food system. For instance, Gao et al. [21] observed that the accumulation of 13S-HODE inhibited the production of OTA in soybean culture medium, but its mechanism is unclear. Understanding of the formation characteristics of oxylipins not only can provide a better understanding of the lipid oxidation process in food, but also enable exploration of their application for the control of fungal growth and mycotoxin production in food. Furthermore, the actual application of oxylipins in foods requires a detailed understanding of its different antifungal and anti-mycotoxin mechanisms, as each oxylipin content may have different action sites. (Lines 314-324)

Reviewer 2 Report

Dear authors

I read your paper thoroughly and have few suggestions for you. All imperfections are of formal character. Please carefully read your whole text and all Latin names turn to italics. Move your tables to the right place. In their current form they are blindly inserted in the text.   

Author Response

Response 1: Thanks for your suggestion. We have revised the manuscript accordingly.

Reviewer 3 Report

Dear Editorial Board of Toxins Journal

Good day;

Great thanks to the authors for their choice of this interesting topic.

and the view of discussion was significant. 

However, I would like to refer to a few points urgently need that turn the review more interesting.

 1 - Introduction could be supported by works of literature that evaluate the impact of oil (crude, pure, micro, nano) forms where it was applied in fungal media growth.

2- For the discussion of the fatty acid / Oxylipin effect either for fungal growth or for toxin production; this point will be more clear if you try to use the flow charts, diagrams, or an imagined form of the reaction changes.

3-  for the title "  Roles of oxylipins in regulating fungal growth and mycotoxin production ", the authors already use an appropriate number of references, but the discussion needs to be more clear.

4 - The conclusion needs moderate changes to elucidate the idea, and the future prospects and recommendations need to be present. 

Author Response

Response 1: Thank you for your suggestion. Oils (especially nano-essential oils) are new strategies to solve the above problems, not only for being natural origin, but also because of having proven preservative efficacy against mycotoxin production. However, the high cost of essential oils and the controversy in biosafety have restricted its practical application [4-6]. (Lines 39-42)

Response 2: The reviewer provided us a good suggestion. We have added a schematic diagram (Figure 1) in the manuscripts accordingly.

Response 3: Thank you for your suggestion. Fatty acids are known to possess antifungal activity. Compared with fatty acid, its derivative - oxylipins have higher reactivity. Oxylipin also can be used as a natural preservative in food to control fungal growth and subsequent mycotoxin production. There is evidence that monohydroxy octadecenoic acid exerts antifungal activity in bread [30]. Similar to FAs, the effect of oxylipins on fungal growth and mycotoxin synthesis also depends on their structure and concentration. Therefore, fungi and mycotoxins in foods can be controlled by adding oxylipins to foods or by modifying processing technology to regulate the formation of certain oxylipins in foods. We added this point into our revised manuscript and the details can be found in Lines 112-116 and Lines 128-130.

Response 4: Thank you for your suggestion. We have readjusted the contents of “Conclusion and remarks” section. For details, see Lines 335-352.